# Improved In Vitro Model for Intranasal Mucosal Drug Delivery: Primary Olfactory and Respiratory Epithelial Cells Compared with the Permanent Nasal Cell Line RPMI 2650

**DOI:** 10.3390/pharmaceutics11080367

**Published:** 2019-08-01

**Authors:** Simone Ladel, Patrick Schlossbauer, Johannes Flamm, Harald Luksch, Boris Mizaikoff, Katharina Schindowski

**Affiliations:** 1Institute of Applied Biotechnology, University of Applied Science Biberach, Hubertus-Liebrecht Straße 35, 88400 Biberach, Germany; 2Institute of Analytical and Bioanalytical Chemistry, University of Ulm, Albert-Einstein-Allee 11, 89081 Ulm, Germany; 3School of Life Sciences, Technical University of Munich, Liesel-Beckmann-Straße 4, 85354 Freising-Weihenstephan, Germany

**Keywords:** barrier model, nose-to-brain, primary cells, RPMI 2650, olfactory epithelium, respiratory epithelium

## Abstract

Background: The epithelial layer of the nasal mucosa is the first barrier for drug permeation during intranasal drug delivery. With increasing interest for intranasal pathways, adequate in vitro models are required. Here, porcine olfactory (OEPC) and respiratory (REPC) primary cells were characterised against the nasal tumour cell line RPMI 2650. Methods: Culture conditions for primary cells from porcine nasal mucosa were optimized and the cells characterised via light microscope, RT-PCR and immunofluorescence. Epithelial barrier function was analysed via transepithelial electrical resistance (TEER), and FITC-dextran was used as model substance for transepithelial permeation. Beating cilia necessary for mucociliary clearance were studied by immunoreactivity against acetylated tubulin. Results: OEPC and REPC barrier models differ in TEER, transepithelial permeation and MUC5AC levels. In contrast, RPMI 2650 displayed lower levels of MUC5AC, cilia markers and TEER, and higher FITC-dextran flux rates. Conclusion: To screen pharmaceutical formulations for intranasal delivery in vitro, translational mucosal models are needed. Here, a novel and comprehensive characterisation of OEPC and REPC against RPMI 2650 is presented. The established primary models display an appropriate model for nasal mucosa with secreted MUC5AC, beating cilia and a functional epithelial barrier, which is suitable for long-term evaluation of sustained release dosage forms.

## 1. Introduction

Drug delivery to the brain for the treatment of central nervous system (CNS)-related diseases became of great interest and one of the most challenging research areas in the last decade [1,2]. The brain has a unique barrier to restrict the entry of neurotoxic substances into the CNS: the blood–brain barrier (BBB). This endothelial barrier, with a low rate of pinocytosis and strong tight junctions, is a major hurdle in CNS drug delivery. Over the last years, the need for alternative drug delivery strategies became more and more obvious [3,4]. One of these strategies is the application of drugs via the intranasal route [5]. In contrast to invasive methods like intraparenchymal, intracerebroventricular, or intrathecal injections/infusions, the noninvasive intranasal approach can be used for chronically given drugs potentially resulting in a higher patient compliance. Further advantages of the intranasal application are the large surface area for absorption, rapid uptake and the avoidance of the hepatic first-pass elimination. Disadvantages of this form of application are the limitation to potent drugs (only small volumes can be applied), the mucociliary clearance, and the physico-chemical features of the mucus layer (low pH, enzymatic degradation) [6,7]. In general, there are three relevant consecutive steps for nose-to-brain drug delivery (N2B): The first step is overcoming the epithelial barrier, the second step is the transport from the mucosa to sites of brain entry and the last step is the transport to other sites in the CNS [7]. In this work, for simplification only the first step is modelled. The nasal mucosa can be divided into four epithelial types: the squamous epithelium, the respiratory epithelium, the transitional epithelium and the olfactory epithelium. Of these epithelial types, the respiratory and the olfactory epithelium are most likely to be sites of substance uptake [8]. The detailed composition of those epithelial types have been extensively studied [9,10,11,12]. Briefly, the olfactory mucosa covers ~10% of the nasal cavity surface in man. It is characterised by a pseudostratified columnar epithelium that is located in the upper dorsal part of the nasal cavity. Olfactory sensory neurons are located in the epithelial layer, with neurites spanning from the nasal cavity into the olfactory bulb in the brain. In addition to these neurons, three further cell types are part of the olfactory epithelial layer: structuring sustentacular cells that function as epithelial supporting cells, tubular type duct cells from Bowman’s glands and basal cells (progenitor cells) [13,14]. Approximately 90% of the human nasal mucosa is covered with the respiratory epithelium. This pseudostratified columnar secretory epithelium consist of goblet cells, ciliated cells, intermediate cells and basal cells [15].

There are two pathways to overcome the epithelial barrier of the respiratory and the olfactory epithelium: the intracellular pathway and the extracellular pathway. Intracellularly transported substances can either be endocytosed into olfactory sensory neurons and subsequently be transported to the brain via their axons, or they can be transported via transcytosis across sustentacular cells into the underlying connective tissue (*lamina propria*) [16,17,18]. The extracellular pathway comprises paracellular diffusion into the *lamina propria* through intercellular clefts [18]. The drug’s pathway through the mucosa mainly depends on its lipophilicity and the molecular weight [19]. For small and low molecular weight hydrophilic molecules, such as fluorescein-labelled isothiocyanate-dextran (FITC-dextran; 4.4 kD) or sodium fluorescein (0.37 kD), mainly the paracellular pathway is reported [20,21,22,23,24,25]. These nontoxic and easily detectable fluorescence labelled chemicals are widely used model substances for drug permeation studies [22,23,26]. In contrast, the transcellular transport is described for large molecules such as proteins [27].

To sum up, the nasal mucosa has become a focus of interest for drug application, to overcome the BBB issue. However, for research the olfactory mucosa as well as the dorsal part of the respiratory mucosa is rather difficult to access [28]. A feasible and simple solution may be the use of *post-mortem* porcine mucosa. The pig’s nasal mucosa resembles well the human nasal histology and physiology [29]. The approach to use pigs as model organism for ex vivo mucosa-related experiments is well established [29,30,31,32,33]. The major problem using ex vivo mucosa explants is the limited lifespan of the tissue even under nutritional support. A promising alternative are in vitro models, using epithelial cells under controlled external experimental conditions, allowing a simplification to display only the first barrier without considering blood flow, systemic distribution and other *lamina propria*-related effects [26]. There are two different approaches available for the establishment of in vitro epithelial barrier models. First, there are standard tumour cell lines like the human nasal epithelial cell line RPMI 2650, and primary epithelial cell models can be cultivated as second option [34,35]. These two cell types vary greatly in their lifespan and their differentiation ability. In theory, tumour cell lines are immortal, whereas primary cells are limited to a certain number of cell divisions [36]. However, most tumour cells lack the ability to fully differentiate into, e.g., ciliated or mucus producing cells, while primary cell cultures are morphologically and physiologically very close to their native state [26,37]. As factors such as cilia and mucus can possibly influence the drug uptake time and the bioavailability due to, e.g., enzymes even in stagnant cell culture, these are important parameters to consider for drug permeation evaluations. This is especially important to evaluate drug potential in vitro for simulations of the whole nasal mucosa as it was recently built up by Na et al. (2017) with commercially available human nasal primary cells [38].

Epithelial cell features such as the presence of cilia, mucus secretion and tight junction formation are important factors that influence the bioavailability of nasally applied drugs [7]. Generally, a heterogenic mixture of different cell types is preferred for a valid model, since this better represents the mucosal epithelial assembly in respect to the permeation profile of drugs through the cellular barrier [37,39]. Furthermore, cultured tumour cells tend to build multilayers, as their growth is mostly not limited by contact inhibition after reaching confluency, whereas primary cells remain in monolayers. For epithelial barrier models, monolayers should be preferred as these resemble the natural epithelial structure more closely. Nevertheless, there are several studies proving the tumour cells line originating from nasal squamous epithelium (septum) RPMI 2650 is a suitable model for the nasal epithelium [20,21,26,37,40]. This cell line is closely related to normal human nasal epithelium in terms of its karyotype and the cytokeratin polypeptide pattern as well as the presence of mucus on the cell surface [34,35,41]. Also the metabolic features of the cellular barrier were shown to be vaguely similar to the one of excised human nasal tissue [42,43]. A special characteristic of nasal epithelial cells is their ability to adapt to growth under air–liquid interface (ALI) conditions. The vast majority of recently developed cellular airway models use this method developed by Tchao et al. (1989) [44]. Hereby, cells are grown on porous plastic membrane inserts in multiwell plates with basolateral medium supply whereas the apical side of the cellular layer is exposed to air. These ALI conditions naturally drive cells to form strong tight junctions and to start differentiating towards, e.g., ciliated or mucus producing cells [45,46,47]. Hereby, the lack of differentiation of tumour cells can be a major problem to mimic epithelial barriers. In an attempt to validate these different approaches, Mercier et al. (2018) postulated generally accepted key criteria that appropriate in vitro barrier cell models should meet [37]. These key criteria comprise a sufficient high transepithelial electrical resistance (TEER) coupled with the presence of specific tight junction proteins such as *zonula occludens* 1 (ZO-1) and adherents junction proteins such as E-cadherin when grown under ALI conditions [48,49]. Furthermore, cellular models must allow a measurement of paracellular permeability and ideally express drug transporters for transcellular permeation. The ability to measure paracellular permeation of small to medium sized model substances like sodium fluorescein or FITC-dextran and the screen for cilia and marker proteins such as ZO-1 and mucins allows a comparison of different models.

In our research group, we focus on region-specific model development and application methods for the olfactory mucosa and for nose-to-brain drug delivery. Flamm et al. recently published a method to deliver drugs directly to the olfactory region in mice [50]. Furthermore, Röhm et al. established a screening platform for aerosolizable protein formulations for intranasal drug delivery [51].

In accordance to former investigations, in this study an epithelial barrier model for intranasal delivery was established and characterised that used olfactory and respiratory primary cells. We considered criteria that strongly affect intranasal delivery such as mucin production and cilia formation important for mucociliary clearance. First, the differences between respiratory and olfactory primary cells were investigated. Second, the primary cell barrier models were evaluated and compared against the well-established tumour cell line RPMI 2650. The suitability and feasibility of primary cells as nasal epithelial barrier models for intranasal delivery studies was determined by immunofluorescence, molecular and biochemical investigations of marker proteins, TEER value determination and FITC-dextran permeation.

## 2. Materials and Methods

### 2.1. Cell Culture

#### 2.1.1. Primary Cells

Primary cells were harvested from mucosal explants from the respiratory and olfactory region of 4–6-month-old slaughterhouse pigs. Respiratory tissue was dissected with a short *post-mortem* delay of below 1.5 h from the *concha nasalis ventralis* (*c.n. ventralis*) and olfactory tissue from the dorsal part of the *concha nasalis dorsalis* (*c.n. dorsalis*) and the *concha nasalis media* (*c.n. media*) (see Figure 1). The dissected mucosa explants were disinfected using Octenisept^®^ (Schülke & Mayr GmbH, Norderstedt, Germany) and washed twice with PBS (cell culture grade, Gibco^®^ Invitrogen, Darmstadt, Germany). The epithelial cells were isolated from the connective tissue by incubation for 1 h at 37 °C with 1.4 mg/mL pronase (Sigma-Aldrich, Taufkirchen, Germany) in EBSS (Gibco^®^ Invitrogen, Darmstadt, Germany), 20 U Penicillin-20 µg Streptomycin (PenStrep (10,000 U), AppliChem, Darmstadt, Germany), and 300 I.U./mg gentamycinsulfate (≥590 I.U./mg, Carl Roth, Karlsruhe, Germany; pronase medium, Table 1). To obtain isolated cells the pronase–mucosa suspension was gently agitated, the liquid was carefully removed, and subsequently centrifuged at 700 rpm for 3 min. The supernatant was discarded and the remaining cells resuspended in appropriate volumes of primary culture adhesion medium (DMEM:F12 (1:1), 20% FBS, 2 mM Gln, 1% NEAA, 20 U Penicillin-20 µg Streptomycin, 300 I.U./mg gentamycinsulfate, Table 1). It should be noted that the cultivation in T175 cell culture flasks was highly inefficient compared to smaller formats.

The cells were seeded in collagen-coated cell culture T flasks (T flasks were incubated in advance with 0.05 mg/mL rat tail collagen solution (Primacyte, Schwerin, Germany) for 24 h at 37 °C) with a cell density of ~10^6^ cells/mL and cultivated at 37 °C, 5% CO_2_ and 95% rH. The medium was changed to primary culture medium (DMEM: F12 (1:1), 10% FBS, 2 mM Gln, 1% NEAA, 20 U Penicillin-20 µg Streptomycin, 300 I.U./mg gentamycinsulfate, Table 1) after 24 h. The exchange was necessary to contain fibroblast growth due to reduced serum concentration. Cells were regularly split by trypsination (Trypsin/EDTA, Biochrom, Berlin, Germany) for 10 min at 80% confluence.

For fibroblast depletion, the culture was incubated with trypsin/EDTA solution for 4 min at 37 °C twice a week. The fact that fibroblasts adherent less strongly to the cell culture flask surface was used to reduce fibroblasts by regular short-time trypsination steps The supernatant containing fibroblasts was discarded whereas the remaining adhered epithelial cells were washed with PBS. Fresh cultivation medium was added.

For seeding in cell culture inserts (ThinCert^TM^, Greiner Bio-one, Frickenhausen, Germany), primary cells were isolated from T-flasks by a two-step trypsination to deplete remaining fibroblasts (Figure 1, IV). The cell culture was first treated with trypsin/EDTA solution for 4 min and the supernatant was removed. Subsequently, an additional trypsination step for 6 min was carried out to remove the epithelial cells. Membrane inserts were collagen coated as described before for the cell culture flasks and seeded with 1 × 10^5^ cells per insert. The cells were cultured submerged for one day in the cell culture inserts. After 24 h, the apical medium was removed to cultivate cells under ALI conditions (260 µL medium per well) for 21 days. Cells were apically washed with 200 µL PBS and medium (260 µL/well) was changed every two days.

#### 2.1.2. RPMI 2650 Cultivation

RPMI 2650 cells were cultivated in RPMI 2650 medium (MEM, 10% FBS, 2 mM Gln, 10 U Penicillin-10 µg Streptomycin, Table 1) at 37 °C, 5% CO_2_ and 95% rH. Cells were regularly split at 80–90% confluency by a 5 min trypsination (trypsin/EDTA) treatment.

For permeation experiments, RPMI 2650 cells with passage numbers below 16 were seeded in cell culture inserts (ThinCert^TM^, Greiner Bio-one, Frickenhausen, Germany) with a density of 1 × 10^5^ cells per insert. After 24 h under submerged conditions, the apical volume was removed and the cells were cultivated under ALI conditions for 21 days. The ALI culture was washed every two days apically with 200 µL prewarmed PBS to remove diffused medium. Furthermore, 260 µL fresh medium was applied basolateral.

#### 2.1.3. TEER Measurement 

To evaluate the integrity of the cell layer the transepithelial electrical resistance measurement was used. Therefore, the cell culture inserts were filled apically with 350 µL MEM without phenol red (Gibco^®^, Invitrogen, Darmstadt, Germany) and basolateral filled with 500 µL MEM. For equilibration, cells were incubated for 20 min at 37 °C and 15 min cooled to room temperature (RT). The TEER value of each cell-covered membrane was determined in triplicates using an EVOM epithelial voltohmmeter and chopstick electrodes (World Precision Instruments, Sarasota, FL, USA). The raw data were processed by a blank subtraction (inserts without cells) and the multiplication by the growth area of the membrane (0.336 cm^2^).

### 2.2. Permeation

To perform the permeation experiments, the medium was changed to 260 µL MEM without phenol red. For primary cells 10% FBS was added to the medium. To analyse fluorescein isothiocyanate-dextran solution (FITC-dextran) permeation, 100 µL of a solution with 500 µg/mL FITC-dextran (Sigma Aldrich, Taufkirchen, Germany) in PBS was applied on the apical surface of the cell layer. The experiments were carried out under normal cell culture conditions at 37 °C, 5% CO_2_ and 95% rH. Permeation was studied for 24 h. A volume of 20 µL was taken as sample from the basolateral compartment at 0.5 h, 1 h, 2 h, 3 h, 4 h, 6 h, 8 h and 12 h. To keep the basolateral volume constant, 20 µL fresh medium was added. The samples were diluted 1:5 with PBS and analysed via fluorescence spectrometry (Spectra Max M5e, Molecular Devices, San Jose, CA, USA) at 490/520 nm.

### 2.3. Immunofluorescence Staining

Immunofluorescence (IF) staining was performed as described previously [31], with the difference that the cell-covered membranes were stained directly in the cell culture insert and transferred to an object slide as last step. Briefly, slides were washed with PBS pH 7.4 three times for 5 min, followed by blocking with block solution (4% BSA, 0.5% Saponin and 10% normal goat serum in PBS pH 7.4) for at least 2 h or overnight. The primary antibodies (Table 2) were diluted 1:100 in blocking buffer without normal goat serum and incubated in the cell culture insert for 24 to 48 h at 4 °C. Subsequently, the slices were washed (5 min, 10 min, and 15 min) and incubated with the adequate secondary antibodies (Table 2, 1:500 diluted in PBS pH 7.4) for 2 h. After washing again three times slides were mounted with Fluoroshield™ mounting medium containing DAPI (4′,6-diamidin-2-phenylindol; Sigma-Aldrich, Taufkirchen, Germany).

### 2.4. Cryosectioning and Colorimetric Staining of Cell Culture Insert Membranes

Cell culture inserts were fixed in 4% paraformaldehyde for 10 min, cryo-conserved in 30% sucrose overnight and stored at 4 °C until sectioning. Before sectioning, the membranes were embedded in gelatine (7.5% gelatine and 30% sucrose in PBS [52]) to allow precise adjustment for sectioning. The membranes were cut in 14 µm slices using a cryostat at −25 °C (HM525 NX, Thermo Fisher Scientific, Dreieich, Germany) and mounted on Superfrost^®^ Plus Micro slides (VWR, Darmstadt, Germany). For hematoxylin-eosin staining, slides were washed with distilled water for 1 min, followed by a 5 min hematoxylin staining step. The slides were destained under running tap water for 3 min, and counterstained with Eosin Y-0.5% acidic acid (Sigma-Aldrich, Taufkirchen, Germany) for 5 min. After an additional destaining step (0.5 min), the slides were dehydrated (75% ethanol, 96% ethanol, 100% ethanol, Xylene; 2 min each) and mounted in Eukitt® Quick hardening mounting medium (Sigma-Aldrich, Taufkirchen, Germany).

### 2.5. Reverse Transcription PCR (RT PCR)

The transcript analysis was carried out as described before [31]. Briefly, the total RNA was isolated from cell lysates using TRIzol (Invitrogen^TM^, Dreieich, Germany) according to the manufacturer’s instructions. For reverse transcription, 1 µg of total RNA was used. Via oligo dT15 primers (Bioron, Ludwigshafen, Germany) and 400 U Reverase^®^ (Bioron, Ludwigshafen, Germany) mRNA was transcribed to cDNA. 

For PCR analysis of the cDNA library 1 µg cDNA, 1 µM primers (see Table 3), 25 mM MgCl_2_, 2.5 mM dNTP Mix and 0.5 U/µL Taq polymerase were diluted in 10× Taq-PCR buffer (all Bioron, Ludwigshafen, Germany).

The PCR reaction was performed with the following parameters: initial denaturation for 30 s at 95 °C, 40 cycles with denaturation at 95 °C for 30 s, annealing at 60 °C for 30 s, elongation at 72 °C for 60 s and a final elongation at 72 °C for 10 min. The PCR products were analysed using agarose gel electrophoresis.

### 2.6. Dot Blot

MUC5AC expression was analysed via dot blot as the high molecular weight protein is difficult to analyse via western blot [53]. Samples were collected from apical washing of ALI cultures with PBS as well as from tissue and cell lysates. The lysates were obtained as described before [31]. For dot blot analysis, 2 µL of these samples were applied onto a nitrocellulose membrane. The membrane was blocked by incubation with 5% skimmed milk powder (in PBS pH 7.4, 0.1% Tween20 (PBST)) for 2 h at RT. The anti-MUC5AC primary antibody (Table 2) was diluted 1:5000 in PBST and incubated for 24 h at 4 °C. The membrane was washed 4 times with PBST, and subsequently incubated with the HRP-coupled secondary antibody (dilution 1:4000 in PBST, Table 2) for 1h at RT. After another 4 washing steps, the membrane was developed by using the chemiluminescence substrate Immobilon^®^ (Merck Millipore, Darmstadt, Germany). The analysis was carried out by Fusion FX Imaging systems (VILBER Lourmat, Collégien, France) and Image J (java.version: 1.8.0_171, National Institute of Health, Bethesda, MD, USA).

### 2.7. Western Blot

The western blot was performed as described before [31]. Briefly, the cell lysates were made by using chilled RIPA cell lysis buffer (10 mM Tris-Cl, pH 8.0; 1 mM EDTA, 0.5 mM EGTA, 1% Triton X-100; 0.1% sodium deoxycholate, 0.1% SDS and 140 mM NaCl; and protease inhibitor mix (Thermo Fisher Scientific, Dreieich, Germany) and gentle agitation. Equal volumes of homogenized tissue were loaded, separated in a 12.5% SDS PAGE and blotted onto a nitrocellulose membrane (Carl Roth, Karlsruhe, Germany). The membrane was blocked (5% skimmed milk powder in PBS/0.1% Tween20, pH 7.4). Primary antibodies (Table 2, 1:5000) were incubated overnight at 4 °C. Secondary antibodies were diluted 1:100,000 (Anti-rabbit IgG-HRP) and 1:4000 (Anti-murine IgG-HRP). Signal detection was carried out by developing the membrane with the chemiluminescence substrate Immobilon^®^ (Merck Millipore, Darmstadt, Germany) according to the manufacturer’s instructions.

### 2.8. Statistics

Data were assessed for significance using unpaired *t*-test (GraphPad Prism 8) comparing the primary nasal cells to the RPMI 2650 cell line. The exact repeat numbers are divided into technical replicates (*n*) and biological replicates (*N*) meaning different pigs and are addressed in the subtitles of the graphs.

## 3. Results

The increasing interest in intranasal delivery requires the development and the evaluation of specialized in vitro models to reduce the use of laboratory animals for drug transport and permeation studies. These models should display the main parameters of the mucosa that influence the drug permeation. Here, we defined the most important parameters as the growth in defined monolayers, the formation of cilia (drug clearance), tightly connected cell layers and the physico-chemical environment (mucus). The actual state of the art model for nasal epithelial cells is the tumour cell line RPMI 2650 [21,39]. As an alternative to tumour cells lines, primary cell models are moving more and more into focus in many fields. Here, the aim of this study was to establish a robust protocol for nasal primary epithelial cell cultivation and to develop an appropriate primary epithelial cell model. This model should especially display the first epithelial barrier during intranasal drug delivery, and hence will be compared to the standard in vitro cellular model.

### 3.1. Evaluation of Nasal Primary Cells and RPMI 2650 Concerning Olfactory Mucosa Model Characteristica-Monolayer, Tight Growth, Cilia and Mucus Production

A general observation of primary nasal cell cultivation was the slow growth of epithelial cells along with faster growth of undesired fibroblasts and the rapid growth of bacterial contaminations (Figure 2A,B). For fibroblast growth we have determined the content of serum in the cell culture medium to be the most influencing factor. Primary cells displayed poor adherence to the collagen coated cell culture flasks at serum levels of 10%. Increasing the serum concentration to 20% resulted in primary cells that were adherent on collagen-treated cell culture flasks after 3–4 h. However, a serum content of 30% increased, above all, the growth of fibroblasts and decreased the growth of epithelial cells. Therefore, despite the positive effects on adhesion to cell flask surface the negative effects on fibroblasts’ growth counterbalanced these benefits. Hence, seeding the cells in 20% serum content followed by a change to 10% serum after adhesion (5 h) turned out to be method of choice to culture the primary porcine nasal epithelial cells.

Morphologically, a majority of respiratory epithelial primary cells (REPC) and olfactory epithelial primary cells (OEPC) appeared in the typical flat and squared cobblestone shape with sizes varying from 10 µm to 100 µm (Figure 2C,D). Particularly in REPC cultures, some cells were detected that contained larger vesicles (Figure 2E). These cells had morphological similarities to goblet or gland cells. Ciliated cells were observed in OEPC and REPC, however REPC showed considerably more cells with motile cilia, whereas in OEPC primary, non-motile cilia were also present (Figure 2F). The motile cilia showed a median length of 10 µm with a beating frequency of >5 Hz (Appendix A).

To investigate the general cell growth of primary nasal epithelial cells and the tumour cell line RPMI 2650 after 21 days under ALI conditions, colorimetric staining (Hematoxylin/Eosin, Figure 3A–C) of cryosections of cell-covered membranes was performed. OEPC and REPC grew in monolayers under ALI conditions whereas the RPMI 2650 cells formed up to 10 layers. In contrast to OEPC, REPC cultures consisted of larger and smaller cells resulting in an uneven monocellular layer surface (Figure 3B). Additionally, the distribution of cilia and the formation of tight junctions were determined using immunofluorescence (IF) staining against acetylated α-tubulin (cilia marker [54]) and ZO-1 (tight junction marker [55]) (Figure 3D–F). OEPC demonstrated the highest signal for the cilia marker acetylated α-tubulin. IF against ZO-1 showed that REPC formed the highest number of tight junctions. Only a weak signal for both marker proteins was observed in RPMI 2650.

Beyond cilia and tight junctions, the physico-chemical environment of the epithelium also has an influence on intranasally administered substances. Therefore, the production of the mucin MUC5AC was investigated using IF (Figure 3G–I), RT-PCR (Figure 4A) and dot blot (Figure 4B). Both the dot blot cell lysate analysis, as well as the IF analysis, showed that OEPC have the highest immunoreactivity against MUC5AC. In the RT PCR no significant difference between *c.n. media* and the nasal primary epithelial cells could be observed. In contrast, the RPMI 2650 cells showed a significantly lower *MUC5AC* transcript level and no signal in the dot blot. Immunolabelling against MUC5AC resulted in weak signals in RPMI 2650 and REPC compared to OEPC.

Furthermore, we analysed the presence of secreted MUC5AC on the apical surface of the OEPC ALI cultures by washing the luminal compartment each 3 to 4 days. The supernatant of the washing buffer was then analysed by dot blot for immunoreactivity against MUC5AC (Figure 4C). At day 6 under ALI conditions the highest signal was observed. After 10 days the protein level appeared to stay at a constant level up to 20 days.

### 3.2. FITC-Dextran Permeation Thought RPMI 2650 and Nasal Primary Cell Barriers

The analysis of the TEER values of 21 days ALI cultures of RPMI 2650, OEPC and REPC ALI revealed that REPC with 846 ± 550 Ω cm^2^ displayed the highest TEER values followed by OEPC (648 ± 371 Ω cm^2^) and last RPMI 2650 (66 ± 5 Ω cm^2^; Figure 5A). Analogous to the permeation data, the nasal primary cells showed considerably higher variability displayed as higher standard deviations compared to RPMI 2650. The highest TEER value measured for OEPC was 1000 Ω cm^2^, whereas REPC reached a TEER value of 1600 Ω cm^2^.

To evaluate cell barrier function, TEER values of the cell layers after 21 days under ALI conditions were determined and permeation of FITC-dextran was analysed from the luminal/apical to the abluminal/basolateral compartment. Determination of permeability was performed over 24 h with different sampling times. The results were compared to a diffusion control without cells (cell culture insert membrane only). Here, RPMI 2650 showed the highest permeability at all times investigated in comparison to OEPC and REPC. After 24 h, 11.7 ± 0.9% of FITC-dextran permeated through RPMI 2650 cells, whereas OEPC showed a permeation of 3.1 ± 2.0% and REPC resulted in 3.7 ± 2.0% FITC-dextran (Figure 5B). An equilibrium is obtained for FITC-dextran in the luminal and abluminal compartment of the insert. It should be noted that this equilibrium establishes earlier in the control insert without cells and limits the maximal flux of molecules in this static system (see Appendix A). As the published literature usually does not acknowledge this fact, we normalized these results to the maximal concentration of FITC-dextran that is able to diffuse to the abluminal compartment within the same time (Appendix A). The molecular flux was calculated for each cell type investigated and displayed as the amount of FITC-dextran that was able to cross the epithelial barrier per time and distance (Figure 5C). Here, it is obvious that RPMI 2650 cells are rather leaky and display a significantly higher flux compared to OEPC and REPC.

In these experiments, we observed a correlation of permeability and the TEER value for OEPC and REPC (Figure 5D). For RPMI 2650 cells no correlation occurred as the measured TEER values showed a very narrow range from 55 Ω cm^2^ up to 75 Ω cm^2^. Hereby, a higher TEER value did not result in a lower flux. In contrast to REPC and OEPC, a higher TEER value resulted in a lower flux. We determined a minimum TEER value of about 300 Ω cm^2^ that appears to be necessary for primary cells to achieve reproducible permeability results. Hence, all cell-layered inserts used for the flux experiments with FITC-dextran had to meet this criterion.

## 4. Discussion

The increasing importance of intranasal drug delivery leads to a growing need for adequate models to test permeation of drugs through the different nasal epithelia. In the olfactory region, sensory neurons located in the epithelium were implicated to be involved in drug transport: either with a direct uptake by dendritic knobs or by epithelial or gland cells, depending highly on the molecule [5,56,57]. To display the pathway and transport of drugs to the brain in vivo studies are necessary where the intracellular transport in neurons can be determined using mucosa explants ex vivo [31,57]. However, studies on nasal olfactory epithelium are harder to perform in vivo and ex vivo as this area is rather inaccessible and difficult to reach [26,58]. In vitro models are robust tools to simulate active or passive transport through the first epithelial barrier without modelling neuronal transport. Nevertheless, in vitro cell systems can be seen as simplified models of the first epithelial barrier. They can be used to study specific transporters for high molecular weight molecules and unspecific diffusion of small molecules [59,60,61]. Nevertheless, considering transport mechanism and cell type specific characteristic it is necessary to set stringent criteria for these cellular models. In the nasal mucosa the epithelial cells form the first barrier for intranasal drug delivery. They are responsible for either uptake or clearance of the administered drugs [6,62]. Parameters that influence the uptake of substances with higher molecular weight are mainly the cell type (e.g., gland cells, sustentacular cells, immune-related microfold cells, etc.) and the expression of specific transport proteins such as aminopeptidases for peptide permeation studies [26,31,42,63]. Furthermore, the drug clearance alongside of nasal mucosal barriers is highly associated with cilia formation and beating [62]. In the olfactory epithelium the cells are described to form non-motile cilia. However, it was demonstrated that so called respiratory mucosa patches exist in the *regio olfactoria* that consist of epithelial cells with beating cilia. Consequently, it is harder to define the clearance rate of the olfactory mucosa only [10]. In addition, for drug stability the physico-chemical environment has to be considered which includes mucus production and the growth under ALI conditions [64]. According to these parameters that influence drug permeation on nasal mucosa, we set the following criteria for the in vitro model: tight epithelial barrier and tight junction formation, cellular monolayer, cilia formation and mucus production.

Standard nasal permeation models are often based on the tumour cell line RPMI 2650. This permanent cell line is derived from an anaplastic squamous cell carcinoma of the nasal septum, which is part of the respiratory region of the nose. Its human origin makes this cell line advantageous in terms of species comparability. Moreover, this cell line was described to mimic the human nasal mucosa and is well investigated in terms of expression of mucus-like material, aminopeptidases, ABC and SLS transporters, cytokeratin patterns, tight junctions and the growth under ALI conditions [26,41,65,66]. The good characterisation makes the RPMI 2650 cell line an established model for intranasal delivery. However, differences in the handling of this cell line have been reported to highly impact growth and protein expression, which may question the reliability of this model [37,67]. Also, we noticed that the permeation rate, as well as the TEER values, depends on the passage count of the RPMI 2650 cells; the higher the passage the lower the TEER value and the higher the permeation rate (unpublished observation). Thus, the nasal mucosa has a heterogeneous composition of several cell types such as ciliated, goblet, columnar and basal cells whereas the RPMI 2650 cell line is only composed of squamous cells which has a limiting impact on the physiology of this model. Furthermore, like most tumour cell lines also the RPMI 2650 cell line suffers from drawbacks such as genetic instability, lack of differentiation and unstable protein expression profiles [37]. The approach that probably mimics best the in vivo situation are ex vivo tissue explants from the olfactory or the respiratory mucosa. Such explants contain all features and the correct cellular composition of the nasal mucosa. However, to take specimen from human olfactory mucosa is associated with a reasonable risk. Yet, porcine nasal tissue is highly similar to the human nasal mucosa and therefore explants are available from slaughterhouse pigs [29]. However, so far there is no cultivation protocol available for long time studies with those tissue samples. We found out 8 h to be the maximum cultivation time before the tissue integrity is highly compromised in histological sections [31]. Consequently, for long-term studies, e.g., to analyse the influence of drug formulations or drug uptake from sustained release, this model is limited.

Hence, we compared the long-term culture of primary nasal cells derived from porcine respiratory and olfactory mucosa with the standard model RPMI 2650 to gain more information about the relevance of potential artefacts due to the use of a cancer cell lines.

### 4.1. Comparison of OEPC and REPC: Differences in Barrier Formation and Marker Protein Expression

Our results show that there are actually differences between OEPC and REPC in terms of tight junction formation and, consequently, in TEER values, as well as in the expression of the secretory mucin MUC5AC. The permeation experiment, especially the sample time points below 8 h, resulted in a lower permeation through REPC in comparison to OEPC. This can also be related to the higher TEER values of REPC as shown in the correlation of TEER values to permeated FITC-dextran (Figure 5A). The TEER values are amongst others influenced by the thickness of the cell layer and the formation of tight junctions [68]. Both primary cell types form only monolayers that are comparable in size to mucosal epithelial cells. Thus, we conclude that REPC form a tighter barrier in comparison to the OEPC, also shown in the immunostaining of the tight junctions (ZO-1). This is in accordance with the common knowledge of the in vivo olfactory and respiratory epithelium as the epithelial cells in the respiratory epithelium are strongly connected via tight junctions, whereas epithelial cells in the olfactory epithelium are sealed to the olfactory sensory neurons neither by tight nor by gap junctions [69]. Thus, our results concerning the ZO-1 signal of the REPC are comparable to the human nasal primary cell model MucilAir^TM^ [40]. In contrast to the respiratory epithelium, olfactory sensory neurons are located in the olfactory epithelium surrounded by sustentacular cells. The olfactory sensory neurons are sealed to those cells and span from the apical surface to the olfactory bulb [9]. Neurogenesis of olfactory neurons is known to be a continuous process to replace lost neurons in the epithelial layer, which leaves a gap in the epithelial layer until the new neuron takes its place [70]. Therefore, it might be possible that the lower number of tight junctions and the lower TEER value seen in the OEPC is correlated to the neurogenesis and the connection between the sustentacular cells and the olfactory sensory neurons. Pezzulo et al. (2010) described TEER values ranging from 700 to 1200 Ω cm^2^ for primary cells originated from the tracheal and bronchial tract, which are similar to the TEER values for REPC in the present study [47]. However, for human nasal epithelial cells TEER values up to 3155 Ω cm^2^ are reported [71,72]. In contrast, ex vivo TEER measurements of excised human nasal mucosa range from 60 to 180 Ω cm^2^ [37]. The discrepancy between the excised mucosa and the primary cells could possibly be explained by damage to the axons of the olfactory sensory neurons during the dissection. These tight junctions are highly dependent on the presence of certain ions. Ca^2+^ ions were reported to be necessary to form tight barriers and a high electrical resistance. It might be possible that the excised mucosa suffers from ion depletion when it is no longer connected to blood flow and therefore shows lower TEER [73]. TEER measurements in rabbit airway epithelium in vivo resulted in a higher TEER value of 260–320 Ω cm^2^ [74]. Besides the preparation and the tight junction formation, the cellular composition of the tissue, e.g., the presence of olfactory sensory neurons, plays an important role in the TEER measurement. Furthermore, gland ducts such as from Bowman’s glands will also lower the TEER value. In the primary culture we did not observe whole glands and neurons. Hence, the higher TEER values could be a result of lacking gland ducts and void neurons.

The ability to form cilia is shared by REPC and OEPC as seen in the immunoreactivity against acetylated α-tubulin. Different groups postulated that acetylated α-tubulin might be a selective marker for primary (non-motile) and motile cilia of polarized cells [75,76,77]. In culture, the cilia had a mean length of 10 µm and beating frequencies of >5 Hz. According to the literature this indicates healthy cilia formation [78,79,80]. In the culture flasks beating cilia were seen in patch-like clusters whereas under ALI conditions the cell layer showed a confluent signal for acetylated α-tubulin. Nevertheless, to further set up valid experiments for drug clearance studies the differentiation of the epithelial cells towards cells with motile cilia, for example by using retinoic acid, should be optimized [81]. Bateman et al. (2013) for example published a porcine tracheal epithelial cell model to investigate virus infection and replication. They used a complex medium containing epidermal growth factor (EGF) for proliferation and retinoic acid to promote ciliogenesis [81]; similar to the present study they obtained ciliated cells. In contrast, those cells grew in multilayers with TEER values increasing up to 12 days to 800 Ω cm^2^ and a subsequent decrease to 250–300 Ω cm^2^ after 18 days. We do not observe this effect as we found TEER values around 850 Ω cm^2^ for REPC and 650 Ω cm^2^ for OEPC after 21 days of ALI cultivation. As we cultured the cells in serum-containing medium without the additional supplementation of EGF or retinoic acid there must be other factors influencing the barrier function that are present in the serum but not in serum-free bronchial epithelial growth medium. As the use of serum-free medium is rather expensive, it is of high interest to further investigate the barrier formation influencing factors present in vivo and in the serum. In contrast to our findings and the findings of Bateman et al. (2013), Delgado-Ortega et al. (2014) reported a leaky barrier model for cells from newborn pig trachea [81,82]. Probably here again the influence of the in vivo cell location and maybe also the age of the donor on the barrier model plays a role. In terms of intranasal drug delivery, using both cell types—REPC and OEPC—for barrier investigations with donor pigs with a similar age can give more detailed information of the permeation in vivo and the most efficient application site in terms of bioavailability.

In terms of mucus expression, the REPC showed lower levels of the secretory mucin MUC5AC expression but transcript levels similar to the OEPC. It has already been described in the literature that transcript expression and the actual protein amount do not always correlate [66]. The reduction of MUC5AC production in OEPC results most probably from lower amounts of glands cells over time as they do not grow under ALI conditions in vivo but in gland formations. A lack of necessary factors such as calcium can be excluded as the medium is regularly exchanged and its composition is constant. The plateau in MUC5AC production after 10 days is possibly due to the fact that also the epithelial cells (sustentacular cells) themselves are capable of producing mucus in vivo [83]. The mucin subtype MUC5AC is known to be mainly expressed in Bowman’s glands of the olfactory mucosa and is therefore a rather poor marker for respiratory epithelial cells [64,84,85]. We screened for this marker protein as we are particularly interested in modelling the *regio olfactoria* for intranasal drug delivery, and therefore the application of drugs at the olfactory epithelium. The absence of MUC5AC does not mean the cells lack mucus production in general, but only this specific mucin type that is known to be highly expressed in the olfactory mucosa [84]. Aust et al. (1997) found MUC5AC to be expressed only in subpopulations of epithelial cells in the human inferior turbinate [86]. Furthermore, recent studies postulated MUC5AC to be expressed in higher amounts in the olfactory mucosa in comparison to the respiratory mucosa which is in accordance with the results of our study. Those data show clear differences in mucin type expression (MUC1, MUC2, MUC5AC and MUC5B) patterns between respiratory and olfactory epithelial cells [87]. Thus, this study supports our hypothesis that the choice of the cellular model should be based on the site of drug application. MUC5AC expression was previously found in human primary nasal epithelial cells after 21 days of cultivation under ALI conditions which is in accordance to our results with 21 day-ALI culture of OEPC [46]. Also the cultivation conditions were comparable despite of the use of serum-containing media, which however had no obvious disadvantageous effect on the porcine primary epithelial culture.

Nevertheless, taken together there are several differences concerning REPC and OEPC that can influence the drug permeation. It thus appears to be advantageous to use the cell type of the area of interest for the drug application.

### 4.2. Primary Cell Model Evaluation: RPMI 2650 vs. Primary Nasal Cell Barrier

The most prominent disadvantages of primary cells are the high variability between different batches and the slow and limited growth in comparison to tumour cell lines. Because of those, and other reasons, the current standard model for nasal mucosal permeation is the tumour cell line RPMI 2650 [26]. The TEER values and permeation rates measured here fit the results of other groups, showing again that the RPMI 2650 is highly reproducible [20,88,89]. Yet, several differences were observed between RPMI 2650 and primary nasal cells that influence the permeation of molecules. Probably the most influencing factor is the lower TEER value of the RPMI 2650 cells. This goes along with the low expression of tight junction marker ZO-1 in comparison with the primary nasal cells. The paracellular permeation of molecules is highly restricted by these tight junctions in the epithelial barrier [49]. There is a high variance in the TEER values yielded for RPMI 2650 under ALI conditions in the literature from the large range observed (41 Ω cm^2^ to 270 Ω cm^2^), which most probably depends on the size of the cellular multilayer and passage number. Compared to primary cells, RPMI 2650 forms leaky multilayered barriers as described before [22,37,39,88,90]. Mercier et al. (2018) described this discrepancy by the strong dependency on culture conditions and the experience of the operators. Therefore TEER values of 70 to 100 Ω cm^2^ are regarded as valid for cultures grown 21 days under ALI conditions [37]. Returning to the above-mentioned criteria for a suitable in vitro model for intranasal drug delivery, the primary nasal epithelial cells demonstrated a 10 to 13 fold higher TEER value and considerable higher signals for the ZO-1 marker protein. However, the TEER values yielded for the RPMI 2650 cells are similar to the TEER values observed in excised human nasal mucosa (TEER value: 60 to 180 Ω cm^2^) as described above. In addition to the possible explanations of the low TEER value of the ex vivo explants discussed above, it is in general rather questionable to compare TEER value of a single-component system such as the homogenous RPMI 2650 model or even the heterogeneous primary cell models with TEER values from multi-component systems like excised nasal mucosa, as the latter contains components such as e.g., glands and neurons that influence TEER values.

In terms of the production of MUC5AC the RPMI 2650 showed the lowest transcript level and very low signal in the immunofluorescence staining. This is in accordance with previous studies from Berger et al. (1999) who reported discrepancies between the transcript expression and the protein expression in tumour cell lines [66]. In their work they state that synthesis, maturation and secretion are most probably regulated post-transcriptionally. In general, RPMI 2650 are mostly undifferentiated, homogenous epithelial cells that originate from squamous epithelium which is part of the respiratory mucosa. Together with the lack of differentiation to form goblet cells, the lower expression of MUC5AC in comparison to OEPC is therefore explainable [20]. However, in current literature the mucus secretion pattern of RPMI 2650 cells shows discrepancies concerning the expression level and the mucin type as previously described by Mercier et al. (2018) [37]. This might be due to different culture conditions or the use of cells with different ages. To fully evaluate the mucin protein expression of the RPMI 2650 model further investigations are required.

In respect to the last criterion—the formation of cilia—our results showed very low and diffuse signals for acetylated tubulin in RPMI 2650 in comparison to well defined and fibril-shaped structures in OEPC and REPC. Several groups described the RPMI 2650′s inability to differentiate and form cilia which we can support in our present study [37,90]. Furthermore, the tendency of RPMI 2650 to form homogenous multilayers is highly disadvantageous when heterogeneous epithelial layers should be simulated. Also, these cells are not polarized as described for epithelial cells but rather undifferentiated round-shaped stapled cells without a growth direction [91]. It is also questionable whether the cells located in the intermediate layers show the same phenotype as the cells grown at the ALI interface. Especially for transport mechanisms when polarized cells are required the tumour cell model is unfavoured whereas the primary cell model sufficiently fits those criteria.

## 5. Conclusions

In the present study, we demonstrate, for the first time, differences between primary epithelial cells from the olfactory and respiratory nasal region. Consequently, we highly recommend using primary cells originated from the equivalent tissue type for model intranasal application of drugs on the olfactory mucosa. In terms of comparability of porcine and human primary epithelial cells we obtained similar results for ciliogenesis, and mucus production as well as similar TEER values [40,71,92]. As the TEER value is the state-of-the-art parameter to evaluate barrier models, the pig is corroborated as an adequate surrogate for human tissue. Furthermore, for the investigation of complex drug transport pathways primary cell culture should be favoured instead of RPMI 2650 tumour cell lines as those cells have different physiological features due to their growth in multilayers and their lack of differentiation ability. In general, we assume the heterogeneity of the primary cells to be advantageous compared to the RPMI 2650 as we see differences in uptake rates of molecules in different cells in our preliminary data. In conclusion, we strongly recommend the observation of several factors such as the analyte molecule, receptor or transporter expression, the importance of adequate physico-chemical environment and the impact of cilia to choose the correct model for the permeation investigation. The primary cell model is a cheap and robust alternative to the tumour cell line RPMI2650 with the advantage of having all features of epithelial cells in vivo and lacking the throwbacks of tumour cells such as genetic instability. It thus has the potential to replace the tumour cell model even in early assessment of drug development as drug efficacy and toxicity can be determined with a higher comparability to the in vivo situation. However, to fully evaluate the full potential of the primary cell model ex vivo experiments should be performed to compare the permeation of a paracellular transported molecule such as FITC-dextran. Furthermore, our study focused on the barrier function and cell type specific features of the cellular models. We only studied the paracellular transport. For further evaluation, the permeation of transcellular transported molecules such as mannitol should also be assessed.

## Figures and Tables

**Figure 1 pharmaceutics-11-00367-f001:**
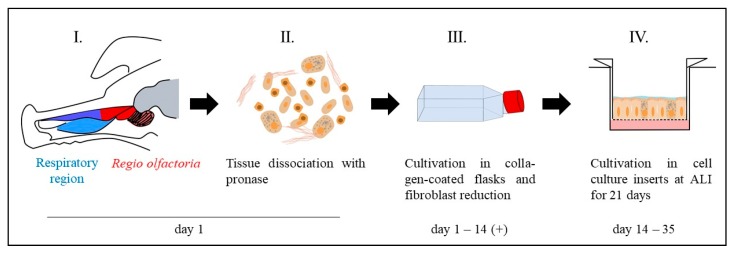
Workflow for primary cell isolation and cultivation under air–liquid interface (ALI) conditions. The mucosa was dissected either from the olfactory region (red) or the respiratory region (blue) (I.). Single cells were obtained by pronase digestion of the mucosa explants (II.). Single cells were cultivated in collagen-coated T75 cell culture flasks (III.). To reduce fibroblast overgrowth, the culture was shortly trypsinated (2–4 min) to get rid of the less adherent fibroblasts and select for epithelial cells. The cells were cultivated up to 80–90% confluence in the T75 flasks and transferred into cell culture inserts to grow under ALI conditions for 21 days (IV.).

**Figure 2 pharmaceutics-11-00367-f002:**
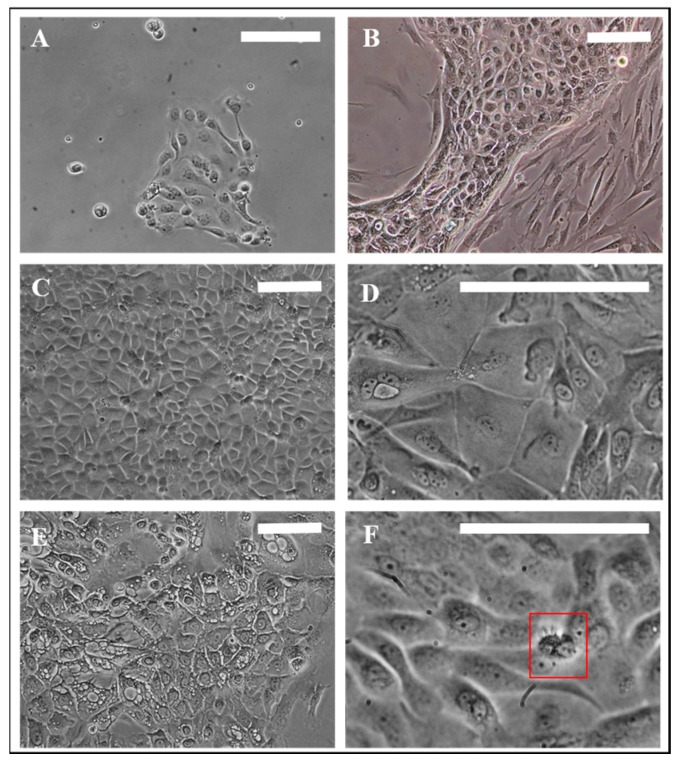
Morphology of porcine primary nasal epithelial cells. (**A**) Olfactory primary cells, 24 h in culture. (**B**) Fibroblast contamination in primary epithelial cells. (**C**,**D**) Morphological appearance of OEPC and REPC: small round or cobblestone shaped and big flat epithelial cells; cell membranes are clearly visible. (**E**) REPC: vesicular cells. (**F**) Red box: Ciliated cells can be motile or non-motile (beating frequency of >5 Hz). Cilia length was ~10 µm. OEPC: olfactory epithelial primary cells; REPC: respiratory epithelial primary cells; Scale bars: 100 µm.

**Figure 3 pharmaceutics-11-00367-f003:**
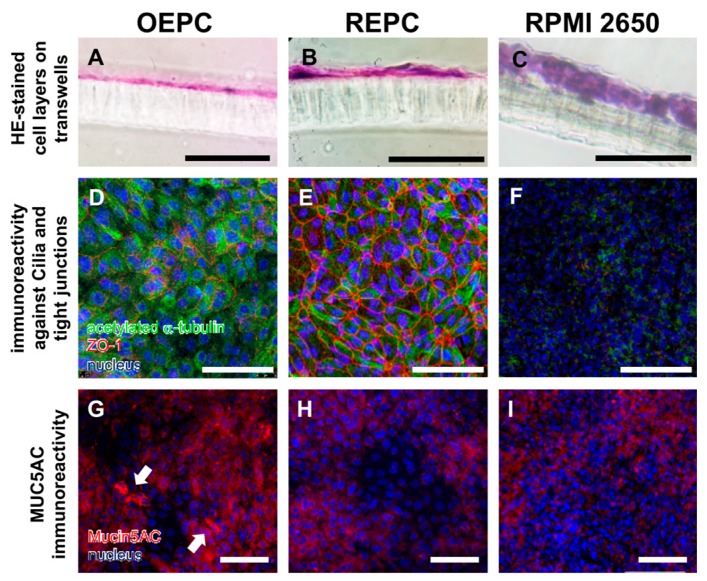
Characterisation of olfactory and respiratory primary cells in comparison to the standard cell line RPMI 2650. Monolayers are necessary to evaluate transport over the epithelial layer in the nasal mucosa: 14 µm sections were made of olfactory epithelium primary cells (OEPC, **A**), respiratory epithelial primary cells (REPC, **B**) and RPMI 2650 (**C**) grown on a cell insert membrane for 21 days. Morphological features such as tight junctions and the formation of cilia are important influencing factors in investigations of drug permeation and clearance studies. Acetylated α-tubulin is a common marker for cilia [54]. IF double-staining of acetylated α-tubulin and the tight junction marker zonula occludens-1 (ZO-1) of ALI cultures of OEPC (**D**), REPC (**E**) and RPMI 2650 (**F**) after 21 days of incubation were made. An additional feature of mucosal cells is the ability to produce mucus. The marker protein mucin 5AC was used in this work, because the olfactory mucosa is to be simulated above all to investigate nose-to-brain transport. Again, IF was performed in OEPC (**G**), REPC (**H**) and RPMI 2650 (**I**). Scale bars: 100 µm.

**Figure 4 pharmaceutics-11-00367-f004:**
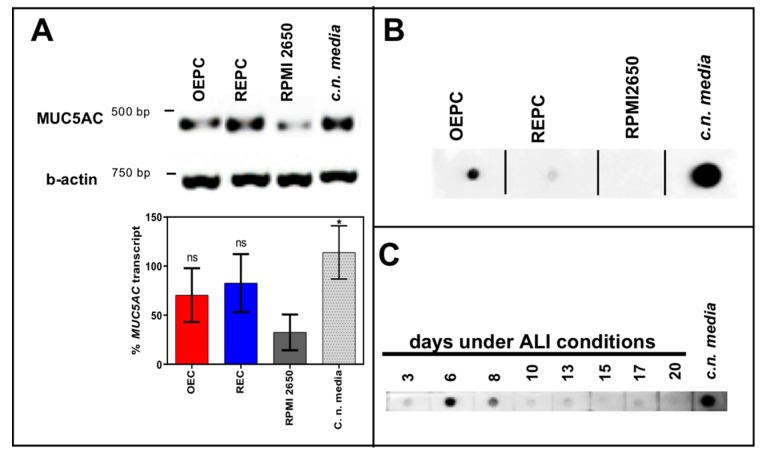
Mucin MUC5AC expression and immunoreactivity in primary cells of the nasal cavity. (**A**) Transcription analysis (RT-PCR) of *MUC5AC* gene in olfactory epithelial primary cells (OEPC), respiratory epithelial primary cells (REPC), tumour cell line RPMI 2650 and the *concha nasalis media* (*c.n. media*). MUC5AC transcript signal was referenced to beta-actin transcript signal. The significance was calculated by comparison of the OEPC, REPC and RPMI 2650 data with the *c.n. media* transcription data using an unpaired t-test. * *p* < 0.05; *n* = 4; error bars represent mean ± SD. (**B**) Dot blot analysis of MUC5AC protein in lysates of OEPC, REPC, RPMI 2650 and *c.n. media*. All OEPC and REPC cultures shown in (**A**,**B**) were cultivated for 14 days in vitro in T flasks. (**C**) Immunoreactivity against MUC5AC in OEPC that were first cultured for 7 days in T flask with a minimum confluency of 70% then under ALI conditions additional 20 days. Apically secreted mucus was collected at the days indicated corresponding to a mucin production of 2 to 3 days. Statistical analysis: unpaired *t*-test, * *p* < 0.05 compared to the standard model RPMI 2650.

**Figure 5 pharmaceutics-11-00367-f005:**
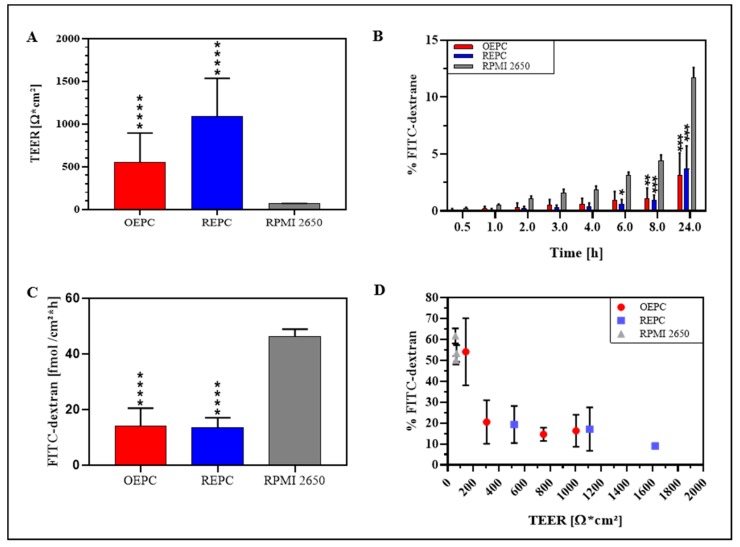
Comparison of FITC-dextran permeation and TEER value of nasal primary cells vs. RPMI 2650. (**A**) TEER values of OEPC, REPC and RPMI 2650 after 21 days ALI. (**B**) FITC-dextran permeation data: Values of FITC-dextran permeated through a cell layer were normalized to total of FITC-dextran. (**C**) Flux of FITC-dextran through an OEPC, REPC and RPMI 2650 layer after 24 h. (**D**) Correlation of percentage FITC-dextran permeation after 24 h and TEER value. Statistical analysis: unpaired *t*-test, * *p* < 0.05, ** *p* < 0.01, *** *p* < 0.001, **** *p* < 0.0001; all compared to the standard model RPMI 2650. OEPC: olfactory epithelial primary cells; REPC: respiratory epithelial primary cells; TEER: Transepithelial electrical resistance. FITC-dextran: Fluorescein isothiocyanate-dextran; *N* = 4, *n* = 21; error bars represent mean ± SD of all repetitions.

**Table 1 pharmaceutics-11-00367-t001:** Composition of the cultivation media used in this work. EBSS = Earle’s balanced salt solution; FBS = Foetal bovine serum; Gln = Glutamine; DMEM = Dulbecco’s modified Eagle’s medium; MEM = minimal essential medium.

Name	Composition
Primary culture adhesion medium	DMEM:F12 (1:1), 20% FBS, 2 mM Gln, 1% NEAA, 0.4 U/mL Penicillin-0.4 µg/mL Streptomycin, 0.6 I.U Gentamycinsulfate
Primary culture medium	DMEM:F12 (1:1), 10% FBS, 2 mM Gln, 1% NEAA, 0.4 U/mL Penicillin-0.4 µg/mL Streptomycin, 0.6 I.U Gentamycinsulfate
Pronase medium	EBSS + 1.4 mg/mL Pronase + 0.4 U/mL Penicillin-0.4 µg/mL Streptomycin, 0.6 I.U Gentamycinsulfate
RPMI 2650 medium	MEM, 10% FBS, 2 mM Gln, 0.4 U/mL Penicillin-0.4 µg/mL Streptomycin

**Table 2 pharmaceutics-11-00367-t002:** List of antibodies used in this study. FITC: fluorescein-isocyanate; HRP: horseradish peroxidase; ZO-1: zonula occludens-1.

Antibody	Antigen	Immunogen	Host	Source, Cat. #
Anti-MUC5AC Antibody (45M1)	Peptide core of gastric mucin M1 (Mucin 5AC)	M1 mucin	mouse	Novus biologicals, Centennial, CO, USA, Cat. #NBP2-15196
Anti-ZO-1 (ZMD. 437)	ZO-1	synthetic peptide derived from the *N*-terminal region of human, dog, mouse, and rat ZO-1	rabbit	Thermo Fisher Scientific, Dreieich, Germany, Cat. #40-2300
Anti-acetylated tubulin (6-11B-1)	Acetylated tubulin	acetylated tubulin from the outer arm of *Strongylocentrotus purpuratus*	mouse	Sigma Aldrich, Taufkirchen, Germany, Cat. #T7451
Anti-β Actin (AC-15)	β Actin	not specified	mouse	Sigma Aldrich, Taufkirchen, Germany, Cat. #A5441
Anti-murine IgG-Alexa Fluor^®^ 488	whole molecule mouse IgG	not specified	Goat	Jackson Immuno Research Europe Ltd., Cambridgeshire, UK, Cat. #115-545-003
Anti-rabbit IgG-Rhodamine Red™-X	whole molecule rabbit IgG	not specified	donkey	Jackson Immuno Research Europe Ltd., Cambridgeshire, UK, Cat. #711-295-152
Anti-murine IgG-HRP	whole molecule mouse IgG	not specified	goat	Sigma Aldrich, Taufkirchen, Germany, Cat. #AP5278
Anti-rabbit IgG-HRP	Whole molecule rabbit IgG	not specified	Goat	Jackson Immuno Research Europe Ltd., Cambridgeshire, UK, Cat. #111-035-003

**Table 3 pharmaceutics-11-00367-t003:** Primer sequences of the targets *MUC5AC* and β-actin for RT-PCR.

mRNA Targets	Forward Primer (5′-3′)	Reverse Primer (5′-3′)
MUC5AC	CCGGGCCTGTGCAACTA	GTTCCCAAACTCGATAGGGC
β-actin	GACACCAGGGCGTGATGG	GCAGCTCGTAGCTCTTCTCC

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
