# Peer review of "Improved In Vitro Model for Intranasal Mucosal Drug Delivery: Primary Olfactory and Respiratory Epithelial Cells Compared with the Permanent Nasal Cell Line RPMI 2650"

_pharmaceutics, 2019, doi:10.3390/pharmaceutics11080367_

Reviewer 1 Report

This manuscript has detailed the development of more biorelevant in vitro cell model as a tool to assess drug delivery via nasal route. The authors have done a fantastic job in the presentation of the manuscript and the experimental methods. Following suggestions and clarifying comments are to improve the manuscript readability.

1. In general, the manuscript is lengthy and large number of references are cited. This is not a major point.

2. At the end of this work, it was suggested to the readers to choose the model appropriately based on advantages and disadvantages of each model. Is the intention of this work to replace the existing RPMI model with the primary cell culture model.

3. To show that the primary culture model is better, it will be better to compare permeability across excised intact mucosa, even if it is for 8 hours. Especially because the authors do not recommend comparing TEER values between in vitro and in vivo scenarios. This will validate the primary culture model.

4. Given the resources involved including cost and time, is the primary cell cculture model suggestible in early assessments of drug development or at later stages? Just curious.

5. The mucociliary action restricts drug delivery in an in vivo situation. It is unclear how the formation of beating cilia would impact drug delivery/permeation in the stagnant cell culture conditions. Please include some text regarding this.

6. In figure 1, may be it will be clear if average number of days are included for each step.

7. Minor typos are seen e.g. line 335, RPMI instead of PRMI

Author Response

Simone Ladel and Katharina Schindowski

Hubertus-Liebrecht StraĂźe 35

88400 Biberach and der RiĂź, Germany

ladel@hochschule-bc.de

zimmermann@hochschule-bc.de

Reviewer 1

BiberachJuly 17, 2019

Dear Sir or Madame, 

thank you for your constructive and detailed review and your valuable comments. 

To your remarks: 

1. We shortened the introduction (line 44-47;83-87;134-138)and the results part (line 329-333; 383-384, 409-411; 412-413) and we removed the following reference: 

• Thorne, R. et al., 2008, doi:10.1016/j.neuroscience.2008.01.013.

• Thorne, R. et al., 2004, doi:10.1016/j.neuroscience.2004.05.029.

• Morrison, E. et al., 1990, doi:10.1002/cne.902970102.

• Chen, Y.et al., 1994doi:10.3109/00016489409126041.

• Moran, D. T.et al., 1982https://doi.org/10.1007/BF01153516

• Gebert,A et al1994, doi:10.1007/BF00306106.

2. We are convinced that the primary cell model provides more and more valid information on drug permeation concerning molecular transport mechanisms and cell type specific mechanisms (mucus interaction of the drug). Furthermore, the interaction between different cell types simulates the in vivo situation more accurately than the clonally related RPMI 2650 cells. However, primary cell models are work intensive in comparison to tumour cell lines and show slow doubling times. Therefore, we recommend to use the primary cell model after a pre-evaluation (cytotoxicity) of a drug substance in the fast-growing and easy to handle tumour cell model. 

3. You are absolutely right that the final evaluation of the primary cell model need the comparison with the excised intact mucosa. We are already working on that topic. However, the intention of this work was to evaluate the potential of the primary cell model against the well-established tumour cell line RPMI 2650. As excised mucosa is not reliably viable after 6 hours, the first step was to compare the cellular models for long-term experiments. 

4. Thank you for your curiosity! The primary cell model is very cost effective. In principle the only compound you need in addition to standard medium is the enzyme pronase (cost 1g 130 â‚¬). This cell model gives a lot of relevant information for later in vivo experiments therefore it is reasonable to use this model in early drug assessments. However, the cells take 2 to 3 weeks to become confluent. We recommend to generate some data concerning the cytotoxicity and stability of the drug in the fast growing tumour cells in advance. 

5. We included some text concerning the importance of mucociliary action in in vitro models (line 97-101)

6. We included the hands-on times in the figure 1. 

7. We corrected the spelling mistakes. 

Thank you again for your helpful annotations for the improvement of this manuscript. 

Cordially,

Simone Ladel and Katharina Schindowski Zimmermann (On behalf of all authors)

Institute of Applied Biotechnology (IAB)

Hubertus-Liebrecht-StraĂźe 35

88400 Biberach an der Riss

Germany

Reviewer 2 Report

Please clarify if the data was presented as mean+/-SD and provide the repeat numbers. 

Shorten the list of references. Lot of articles were published before the year of 2000.

Author Response

Simone Ladel and Katharina Schindowski

Hubertus-Liebrecht StraĂźe 35

88400 Biberach and der RiĂź, Germany

ladel@hochschule-bc.de

zimmermann@hochschule-bc.de

Reviewer 2

BiberachJuly 17, 2019

Dear Sir or Madame, 

Thank you for your constructive and detailed review and your valuable comments. 

To your remarks: 

1. We provided the exact repeat numbers in the figure legends and inserted a short explanation in chapter 2.7. 

2. We removed the following reference: 

• Thorne, R. et al., 2008, doi:10.1016/j.neuroscience.2008.01.013.

• Thorne, R. et al., 2004, doi:10.1016/j.neuroscience.2004.05.029.

• Morrison, E. et al., 1990, doi:10.1002/cne.902970102.

• Chen, Y.et al., 1994doi:10.3109/00016489409126041.

• Moran, D. T.et al., 1982https://doi.org/10.1007/BF01153516

• Gebert,A et al1994, doi:10.1007/BF00306106.

Due to the scientific importance could not remove all references that were published before the year of 2000 as we are need them to support key messages (e.g. characterization of the RPMI 2650 model) and we are concerned to lower the scientific quality of the manuscript.

Thank you again for your helpful annotations for the improvement of this manuscript. 

Cordially,

Simone Ladel and Katharina Schindowski Zimmermann (On behalf of all authors)

Institute of Applied Biotechnology (IAB)

Hubertus-Liebrecht-StraĂźe 35

88400 Biberach an der Riss

Germany

Reviewer 3 Report

The paper, entitled “Improved in vitro model for intranasal mucosal drug delivery – primary olfactory and respiratory epithelial cells compared with the permanent nasal cell line RPMI 2650” describes and characterize a new model based on primary porcin olfactory and  respiratory cells, comparing it with the standard human tumor cell line, RPMI2650.

As the intranasal administration is a mainstream of the current investigations performed to treat central and neurodegenerative diseases, the paper presented by Ladel et al has a high level of novelty.  The document is supported by recent bibliographic references, corroborating the novelty of its scope.

Methodology is well described and a wide diversity of methods were applied, increasing the merit of the work. The results are also well presented and discussed.

As a general comment, text length should be strongly reduced, particularly the introduction and results, which should be stick to the point of the article.

I would suggest the authors to compare their model with in vitro human nasal mucosa models that are reliable for drug screening and clinical applications, such as the 3D MucilAir™ nasal model and microfluid chips and primary human cells which were recently published (DOI: 10.1016/j.ejpb.2019.04.002 and 10.1039/c6lc01564f and 10.1016/j.envpol.2019.02.082 ). 

In addition, I have some minor suggestions to improve paper’s quality

1.       Page 2, line 52: “adsorption” should be replaced by “absorption”

2.       Page 2, lines 61-62, “the respiratory and the olfactory epithelium are most likely to be sites of absorption”. Absorption considers the passage from administration site to systemic blood. The olfactory epithelium is mainly involved in the direct passage into olfactory bulb. Please, correct the phrase.

3.       Section 2.3. should be more specified.  

Author Response

Simone Ladel and Katharina Schindowski

Hubertus-Liebrecht StraĂźe 35

88400 Biberach and der RiĂź, Germany

ladel@hochschule-bc.de

zimmermann@hochschule-bc.de

Reviewer 3

BiberachJuly 17, 2019

Dear Sir or Madame, 

Thank you for your constructive and detailed review and your valuable comments. 

To your remarks: 

1. We shortened the introduction (line 44-47;83-87;134-138) and the results part (line 329-333; 383-384, 409-411; 412-413) and we removed the following reference: 

• Thorne, R. et al., 2008, doi:10.1016/j.neuroscience.2008.01.013.

• Thorne, R. et al., 2004, doi:10.1016/j.neuroscience.2004.05.029.

• Morrison, E. et al., 1990, doi:10.1002/cne.902970102.

• Chen, Y.et al., 1994doi:10.3109/00016489409126041.

• Moran, D. T.et al., 1982https://doi.org/10.1007/BF01153516

• Gebert,A et al1994, doi:10.1007/BF00306106.

2. Thank you for your suggestion of relevant literature concerning primary human cell models, nasal models and microfluidic chips. We integrated and addressed them in our manuscript (line 624, 505, 516). 

Also, all your minor suggestions have been acknowledged.

Thank you again for your helpful annotations for the improvement of this manuscript. 

Cordially,

Simone Ladel and Katharina Schindowski Zimmermann (On behalf of all authors)

Institute of Applied Biotechnology (IAB)

Hubertus-Liebrecht-StraĂźe 35

88400 Biberach an der Riss

Germany

Reviewer 4 Report

This manuscript describes the isolation of two primary cells from the olfactory region and the respiratory region mucosa of pigs to establish two epithelial cell barrier models in vitro, OEPC (olfactory epithelial primary cells) and REPC (respiratory epithelial primary cells). The primary cell barrier models were evaluated and compared against the tumor cell line RPMI 2650 in terms of the growth in defined monolayers, the formation of cilia (drug clearance), tightly connected cell layers and the physic-chemical environment (mucus). It provides a good strategy for studying in vitro models of nasal drug delivery. The manuscript is of novelty, and its related studies and data are clear and convincing. Therefore, it can be accepted after the following comments addressed. 1. In Figure 4A &B, the REPC showed lower levels of the secretory mucin MUC5AC expression but the RT-PCR level similar to the OEPC. In the Figure 4C, MUC5AC expression was increased and then decreased to a stable trend. What factors lead to the change of MUC5AC expression in OEPC? 2. Figure 5A showed the TEER values of OEPC, REPC and RPMI 2650 after 21 days ALI. It is insufficient to only show the TEER value of the last day. If the change of the TEER value within 21 days can be displayed, the growth of the primary cells can be more comprehensively evaluated. 3. The differences of OEPC, REPC and RPMI in the transport of FITC-dextran, a paracellular transport marker, were evaluated. However, some substances are transported by means of trans-cellular pathway. If a transcellular transport marker could be selected for transport evaluation, the difference in transport of three models will be more complete. 4. There are reviews about brain targeting drug delivery (Acta Pharm Sin B, 2016, 6(4): 268-286), authors should refer them.

Author Response

Simone Ladel and Katharina Schindowski

Hubertus-Liebrecht StraĂźe 35

88400 Biberach and der RiĂź, Germany

ladel@hochschule-bc.de

zimmermann@hochschule-bc.de

Reviewer 4

BiberachJuly 17, 2019

Dear Sir or Madame, 

Thank you for your constructive and detailed review and your valuable comments. 

To your remarks: 

1. We addressed your question in line 543 to 550. Shortly, it is known that the transcript and the expression level do not always correlate regarding the mucin production(Berger, J. T et al., 1999 doi:10.1165/ajrcmb.20.3.3383). We believe that the reduction of the MUC5AC expression over time in OEPC is due to a selection towards epithelial cells. Gland cells probably dye over time under ALI conditions. However, it is known that epithelial (sustentacular) cells are capable of producing mucus but in lower amounts than gland cells. 

2. We agree with you that an evaluation of the TEER value over the whole ALI period would be very good data to evaluate the growth of the primary cells. Unforunately, we do not have the possibility to measure the TEER value under sterile conditions so far. We apologize for not being able to provide this data. 

3. Thank you for your advice to include also transcellular markers. Work is in progress at the moment measuring the permeation of transcellular transported proteins. However, in the present study we evaluate the model especially in terms of paracellular transport as the focus was on the barrier function of the model. 

4. Thank you for your suggestion of relevant review. We integrated and addressed it in our manuscript (line 45). 

Thank you again for your helpful annotations for the improvement of this manuscript. 

Cordially,

Simone Ladel and Katharina Schindowski Zimmermann (On behalf of all authors)

Institute of Applied Biotechnology (IAB)

Hubertus-Liebrecht-StraĂźe 35

88400 Biberach an der Riss

Germany